# ANN-Assisted Beampattern Optimization of Semi-Coprime Array for Beam-Steering Applications

**DOI:** 10.3390/s24227260

**Published:** 2024-11-13

**Authors:** Waseem Khan, Saleem Shahid, Ali Naeem Chaudhry, Ahsan Sarwar Rana

**Affiliations:** 1Department of Electrical and Computer Engineering, Air University, Islamabad 44000, Pakistan; waseem.khan@au.edu.pk (W.K.); chaudhryalinaeem@gmail.com (A.N.C.); ahsan.sarwar@au.edu.pk (A.S.R.); 2Institute of Microwave and Photonic Engineering, Graz University of Technology, 8010 Graz, Austria

**Keywords:** beampattern, linear antenna array, semi-coprime array, split-aperture array, staggered steering, artificial neural network, beamwidth, sidelobes, Chebyshev weights

## Abstract

In this paper, an artificial neural network (ANN) has been proposed to estimate the required values of the adjustable parameters of a Semi-Coprime array with staggered steering (SCASS), which was proposed recently. By adjusting the amount of staggering and the sidelobe attenuation (SLA) factor of Chebyshev weights, SCASS can promise a quite small half-power beamwidth (HPBW) and a high peak-to-side-lobe ratio (PLSR), even when the beam is steered away from broadside direction. However, HPBW and PSLR cannot be improved simultaneously. There is always a trade-off between the two performance metrics. Therefore, in this paper, a mechanism has been introduced to minimize HPBW for a desired PSLR. The proposed ANN takes the array of architectural parameters, the required steering angle, and the desired performance metric, i.e., PSLR, as input and suggests the values of the adjustable parameters, which can promise the minimum HPBW for the desired PSLR and steering angle. To train the ANN, we have developed a dataset in Matlab by calculating HPBW and PSLR from the beampattern generated for a large number of combinations of all the variable parameters. It has been shown in this work that the trained ANN can suggest the optimum values of the adjustable parameters that promise the minimum HPBW for the given steering angle, PSLR, and array architectural parameters. The trained ANN can suggest the required adjustable parameters for the desired performance with mean absolute error within just 0.83%.

## 1. Introduction

Reducing HPBW and increasing the PSLR of an antenna or antenna system has been a popular area of research for more than two decades. Researchers have tried to optimize the two performance metrics, either by adjusting the array shape and geometry including inter-element spacing [1,2,3,4,5] and the relative pattern of the excitation currents of elements [6,7], or by optimizing the antenna elements [8,9,10]. A major part of the past research has focused on a single-aperture array in which a set of antenna elements is considered a single array. However, in the last decade, Coprime Sensing Array (CSA) [11] and Semi-Coprime Array (SCA) [12] were proposed, in which an array of sensors or antenna elements is split into subarrays such that the subarrays have few elements common. The beampatterns of the subarrays are combined to form a single beampattern, using different techniques, such as minimum [13,14] or product [15] of the beampatterns or a combination of minimum and product [16].

Investigation into CSA and SCA was triggered by the quest of having a beampattern with smaller half-power beamwidth (HPBW) and higher peak-to-side-lobe ratio (PSLR) with fewer sensors, as compared to the conventional linear arrays. In the initial works, all the subarrays of CSA and SCA have been beam-steered toward the same direction, i.e., the direction of interest. On the contrary, recently, staggered steering of the subarrays has been proposed [17] in which each sub-array is beam-steered in a different direction. If the desired steering angle is $\theta_{0}$, the subarrays are steered towards $\theta_{0}$ + Δ and $\theta_{0}$−Δ, where Δ is the adjustable parameter to control the staggering. Along with staggered beam-steering, in this work, Chebyshev weights were employed to suppress the sidelobes. It was shown in this work that the proposed strategy can outperform the existing split-aperture and unified-aperture arrays in terms of HPBW, PSLR, and directivity if the staggering parameter Δ and the sidelobe attenuation (SLA) of Chebyshev weights are controlled appropriately. However, to control the two parameters, Δ and SLA, no mechanism was devised in [17]. However, it was shown that an increase in Δ results in improving HPBW but may worsen PSLR simultaneously. Similarly, an increase in SLA can improve PSLR but degrade HPBW simultaneously. The results shown in [17] also reveal that the performance metrics HPBW and PSLR have no linear relationship with the adjustable parameters Δ and SLA. Therefore, there is no analytical method to find the values of these parameters that can ensure a given set of performance metrics. However, such problems can be solved using machine-learning techniques such as artificial neural networks (ANNs), which are very good at capturing the non-linear relationships between inputs and outputs.

In past, ANNs have been employed in the domain of antenna arrays for various objectives such as optimizing antenna arrays [18,19,20] and their beampattern [21,22,23], robust beamforming [24], mutual coupling reduction in antenna arrays [25], identification of defective elements of the arrays [26,27], estimation of direction of arrival [28], etc. In this paper, we employed ANN to find a suitable combination of Δ and SLA to ensure optimum HPBW for the desired PSLR for a given steering angle. To enable the ANN, it has been trained on a self-generated dataset. ANNs with different numbers of layers and numbers of neurons in each layer have been trained and evaluated for performance/estimation accuracy. Based on this analysis, two networks, with different numbers of hidden layers and neurons, have been selected; one for the best performance and the other for the best trade-off between performance and computational cost. The trained networks have also been tested on the input feature values not in the training dataset to assess the capability of the network to interpolate and extrapolate.

In this work, the effect of mutual coupling is not considered among the antenna elements. The coupling effect of nearby elements is very critical, and usually, the spacing between the elements is kept at λ/2, to reduce the coupling effect [29]. Increasing the spacing by more than λ/2 has very little effect in reducing the coupling, but it increases the size of the array. Since the minimum inter-element spacing in the proposed work is λ/2, we have assumed that mutual coupling is negligibly small. This work is proposed for beam-steering applications, and the design works best for a single frequency. Therefore, it cannot be used for frequencies far apart. However, in order to work at different close-by frequencies, we can use stretchable, temperature-dependent, or voltage-dependent dielectric materials whose relative permittivity changes with variation in material length, temperature, or voltage, respectively [30]. This may allow necessary changes in inter-element spacing in the array to effectively work at multiple frequencies.

The rest of the paper is organized as follows. Section 2 gives an overview of SCASS and illustrates the objective problem of the adjustment of Δ and SLA that has been addressed in this paper. Section 3 gives an overview of ANNs, elaborates on the dataset generation, and describes the architecture and training of the proposed ANN. Section 4 presents and discusses the results, and finally, Section 5 concludes the paper.

## 2. An Overview of SCASS and the Associated Challenges

The concept of SCASS is based on the fundamental architecture of SCA proposed in [12] and reproduced in Figure 1 for the reader’s convenience. SCA comprises three subarrays (SAs) known as SA1, SA2, and SA3, having PM, PN, and Q elements with inter-element spacing QNλ/2, QMλ/2, and λ/2, respectively. Here, P, M, N, and Q are integers, and M and N are coprime. Physically, it is a single linear array of PM+PN+Q−P−1 elements, spaced non-uniformly, as depicted at the bottom of Figure 1. However, its elements are grouped into three, just for beamforming, so that three different beampatterns can be generated. It is noteworthy here that there is only one element common in the three subarrays. On the other hand, in SAs 1 and 2, there are *P* elements common. This is a sample arrangement of SCA with a specific combination of M,N,P,Q; however, other combinations have also been considered in this paper, as mentioned in Section 3.2. In SCA, the three subarrays are beam-steered towards $\theta_{0}$, which is the desired steering angle of the whole array. On the contrary, in SCASS, a modified version of SCA, the three subarrays are beam-steered towards $\theta_{0}$ + Δ, $\theta_{0}-\Delta$ and $\theta_{0}$, respectively, where $\theta_{0}$ is the desired steering angle of the whole array and Δ is an adjustable parameter to control the deviation of steering angles of SAs 1 and 2. The overall beam pattern of SCASS is given by
(1)$$B\left(\theta \right)=\min\left(B_{1}\left(\theta \right),B_{2}\left(\theta \right),B_{3}\left(\theta \right)\right)$$
where $B_{i}\left(\theta \right)$ is the beam pattern of *i*th sub-array. SCASS has been investigated with uniform (SCASS-U) as well as Chebyshev (SCASS-C) weights. In SCASS-C, Chebyshev weights are applied on SAs 1 and 2, while uniform weights are applied on SA3 [17]. In this paper, only SCASS-C has been considered. Therefore, from this point onward, we shall use the term SCASS to refer to SCASS-C. As shown in [17], SCASS is optimized in terms of HPBW and PSLR when the values of the SLA factor of Chebyshev weights and Δ are selected appropriately. It has also been shown that SLA and Δ can be adjusted to achieve an HPBW smaller than other linear arrays along with a reasonably high PSLR. However, the two performance metrics cannot be improved simultaneously by adjusting SLA and Δ. Improving HPBW results in worsening PSLR and vice versa. To have a clear picture of the effect of the variables Δ and SLA on the performance metrics, we have simulated SCASS to augment the results presented in [17]. We have simulated SCASS, in Matlab, with (M,N,P,Q) = (3, 2, 3, 3) for all combinations of $\theta_{0}\in \left\{0{}^{\circ},30{}^{\circ},60{}^{\circ}\right\}$, Δ∈{0.5°,1°} and SLA ∈{20 dB, 23 dB}. The simulation results have been plotted in Figure 2.

This figure clearly shows the non-linear relationship of HPBW and PSLR with Δ and SLA. It is also obvious from the figure that an increase in PSLR by adjusting SLA is accompanied by increased HPBW. Therefore, it is not possible to attain the smallest possible HPBW along with the highest possible PSLR. In this situation, in this work, we have focused on minimizing the HPBW for the desired PSLR for a given set of M,N,P,Q, $\theta_{0}$. For this purpose, we have implemented and trained an ANN that takes M,N,P,Q, $\theta_{0}$, PSLR as inputs and suggests, as output, a combination of SLA and Δ that can promise the minimum HPBW for the given set of values of the input parameters. Details of the ANN have been discussed in the following section.

## 3. Artificial Neural Network: Introduction and Application to the Problem

In this section, we first introduce the basic architecture of an ANN, then describe the self-generated dataset which has been used to train the proposed network. At the end of this section, we illustrate the architectural configuration and training of the network.

### 3.1. Basic Architecture of an ANN

An artificial neural network (ANN) is a machine-learning model inspired by the human brain that can be trained to perform tasks such as classification and regression [31]. Similar to a human brain, an ANN consists of a network of neurons where each neuron receives multiple inputs and generates an output that may act as one of the inputs to other neurons. An artificial neuron is a mathematical function that calculates the weighted sum of the inputs and adds an offset, also known as bias. Then, it passes the calculated sum to a function, mostly non-linear, usually known as an activation function. An ANN is comprised of an input layer of neurons, an output layer, and one or more hidden layers between them [32], as shown in Figure 3. The number of neurons in the input layer is equal to the inputs (N_*i*_), whereas the number of neurons in the output layer is equal to the number of outputs (N_*o*_). In general, the output of *n*th neuron in *m*th layer is expressed mathematically as follows:(2)$$y_{m,n}=g\left(w_{m,n}^{T}x_{m-1}+b_{m,n}\right),\;n=1,\ldots ,K_{m},\;m=2,\ldots U$$
where $w_{m,n}$ and $b_{m,n}$ are the weight vector and the bias associated with the *n*th neuron of *m*th layer, respectively, $x_{m-1}$ is the output vector of (m−1)th layer, which is also the input of each of the neuron of *m*th layer. Here, $K_{m}$ is the number of neurons in the *m*th layer, *U* is the number of layers, and g(.) is the activation function, which may be sigmoid, linear, ReLU, etc. Please note that the input of the first layer is the input of the network. Therefore, $K_{1}$ = N_*i*_ and K_*U*_ = N_*o*_. According to inter-layer connectivity, ANN can be categorized into fully and partially connected networks. A network consisting of all possible connections among the neurons of adjacent layers is termed a fully connected network (FCN), and a network consisting of a subset of the possible connections is called a partially connected network (PCN). A neural network with a large number of neurons in the hidden layers allows it to learn complex features and patterns. However, a large number of neurons means high computational costs and the risk of overfitting the training data rather than generalizing it.

An ANN is trained by feeding it input values from a set of training examples in which the actual output values are known. During training, the working of ANN involves two processes, namely forward propagation and back propagation. In forward propagation, the inputs from the training dataset cross the input layer into the hidden layers, where each neuron processes the inputs and produces an output that represents an insight, pattern, or feature of the inputs. This array of features then moves to the subsequent layers that further process these features to calculate rather more complex features and nuanced patterns in the inputs, and that is what makes the neural networks strong. Finally, an output is generated for each training example as the output layer processes the manipulated inputs. The discrepancy between the estimated output values, calculated by the ANN, and the actual output values is calculated using a loss function, e.g., mean square error (MSE), mean absolute error (MAE), Huber loss function [33], etc. To train the ANN to give more accurate estimates, back propagation algorithm is employed in which the gradient of the loss function with respect to each weight in the network is calculated. This tells us how much the loss will be affected if a particular weight is changed slightly. Each of the weight vectors is then updated iteratively according to the following relationship:(3)$$w_{m,n}^{i+1}=w_{m,n}^{i}-\alpha \frac{\partial L}{\partial w_{m,n}}$$
where *L* represents the loss function, *i* indicates the iteration number and α is the learning rate. The output of the network depends not only on the weights in the output layer but also on the weights of the neurons in input and hidden layers, to which they are not directly connected. Therefore, the gradient of the loss function is calculated using the chain rule, which allows us to backtrack and see how each weight affects the overall error. Further details about back propagation can be found in [32]. The loss function is minimized iteratively by adjusting the weights and biases of the neurons, using optimization algorithms such as the gradient descent algorithm. There are different types of neural networks, each specialized to handle specific tasks. For instance, Convolutional Neural Networks (CNNs) are used for image recognition, whereas Recurrent Neural Networks (RNNs) handle sequential data, such as text, well [34].

### 3.2. The Dataset Generation

As mentioned in Section 2, the proposed ANN takes the parameters M,N,P,Q, $\theta_{0}$, PSLR as inputs and suggests, as output, a suitable combination of Δ and SLA to ensure the minimum HPBW for the given set of parameters. To train the network, a dataset was generated in Matlab, considering several possible combinations of input values. Four combinations of M,N,P,Q were considered, and $\theta_{0}$, Δ, and SLA were swept in a range with fixed step size, as mentioned in Table 1.

In Matlab, the beam pattern was generated for each set of M,N,P,Q for all combinations of $\theta_{0}$, Δ. SLA and the resulting HPBW and PSLR were noted. The dataset generated after this process contained all combinations of SLA and Δ, including ones not offering minimum HPBW for a specific PSLR. Therefore, in the second step, the dataset was refined by eliminating the training examples with HPBW more than the minimum possible for a specific PSLR. To illustrate the refining process, a subset of raw training examples is shown in Table 2. This table shows that for (M,N,P,Q) = (3, 2, 3, 3), and a specific $\theta_{0}$, there are multiple training examples in which PSLR is the same, i.e., 21.29 dB but HPBW is different. Since our objective is to train the network to suggest the adjustable parameters for minimum HPBW, we discarded all the entries in the dataset in which HPBW was more than minimum for a specific set of M,N,P,Q, $\theta_{0}$, PSLR. As an example, highlighted in the table, HPBW is minimum in the 4th row, among all the entries with PSLR = 21.29 dB for ($M,N,P,Q,\theta_{0}$) = (3, 2, 3, 3, 14°). Similarly, minimum HPBW for ($M,N,P,Q,\theta_{0}$) = (3, 2, 3, 3, 14.5°) is found in the last row for the same PSLR. Hence, the 4th and the last rows were retained, and the rest were discarded. A similar process was repeated for PSLR ranging from 17 to 25 dB for all combinations of M,N,P,Q,$\theta_{0}$. The refined version of the dataset enables the ANN, after training, to suggest a suitable combination of Δ and SLA to offer the minimum possible HPBW for a specified PSLR for a given set of M, N, P, Q, $\theta_{0}$. The trained network also gives the estimated value of HPBW, to be achieved in practice, if the suggested values of Δ and SLA are used for the given set of M, N, P, Q, $\theta_{0}$.

### 3.3. Training and Architectural Configuration of the Proposed ANN

Choosing the architecture of the neural network for a specific application is an iterative process with different combinations of hyperparameters. Generally, a complex mapping of the inputs to the outputs may require a deeper or wider network relative to a simple mapping. It is often necessary to fit a shallow neural network due to its low time and space complexity, whereas deep neural networks have their own set of challenges, such as exploding and vanishing gradients, high computational complexity, training instability, and overfitting [34]. Therefore, it is advisable to begin with a simpler network and gradually increase the complexity while trying a range of hyperparameters like learning rate, regularization coefficients, batch size, etc. After that, a network with the best trade-off between performance and complexity is chosen. In this work, the performance of the network with different numbers of layers and numbers of neurons per layer is depicted in Figure 4. This figure shows that ANN with 3 hidden layers and 32 neurons per layer performed the best, while the ANN with a single hidden layer and 8 neurons per layer performed the worst. However, other ANNs with 1 and 2 hidden layers and 16, 24, and 32 neurons per layer performed slightly poorer than the best one. Now, the selection of the network model depends on whether performance is optimized or the computational cost. If it is desirable to minimize the computational cost with reasonable performance, then the ANN with 1 hidden layer and 16 neurons per layer, known as ANN1, is the best option. However, other specifications of the two networks are shown in Table 3.

The dataset included almost 500,000 training examples split into roughly 70% training set, 20% test set, and 10% validation set. The final model’s training phase involved a typical batch size of 128, an Adam optimization algorithm with default parameters, and a learning rate of 0.1 decayed by 5% after every epoch for a total of 100 epochs. Mean Absolute Error (MAE) was employed as the loss function during the training, which is given by [35]:(4)$$\mathrm{MAE}=\frac{1}{T}\sum_{i=1}^{T}y_{i}-{\hat{y}}_{i}$$
where $y_{i}$ and ${\hat{y}}_{i}$ are the actual and the ANN-estimated output values of *i*th training example, respectively, and *T* is the number of training examples.

## 4. Results and Discussion

The ANN described in Section 3 has been trained on the refined dataset discussed in Section 3.2. The trained network has been then fed with a test dataset carefully generated in Matlab. This dataset has been generated in the same way as the training dataset was, but the values of $\theta_{0}$ chosen in this process are not included in the training dataset. This helps to assess the capability of the network to interpolate. A sample of the test examples is shown in Table 4.

The ANN-estimated values of Δ, SLA, and HPBW have been plotted versus their actual values in Figure 5 for comparison. It is obvious from the figure that the ground truth and the estimated values are more or less the same. The performance of the two networks, in terms of mean absolute error of the three estimated parameters, is also shown in Table 5. The results show that the largest mean absolute error offered by ANNs 1 and 2 are 1.82% and 0.83%, respectively. It is noteworthy here that the proposed ANNs can suggest optimum Δ and SLA only for the reasonable PSLR input values. If the desired PSLR is beyond the capability of SCASS, the ANN-suggested values of Δ and SLA may not guarantee the desired performance. The maximum PSLR that SCASS can offer depends on the array parameters and steering angle, as is obvious from Figure 2. An increase in SLA means more suppression of sidelobes, but it does not necessarily mean increased PSLR. In fact, PSLR increases with an increase in SLA up to a certain limit, which depends on array architectural parameters M,N,P,Q as well as on Δ and $\theta_{0}$. For all the combinations of M,N,P,Q, considered in this work, the highest achievable PSLR is 23.5 dB, which is quite reasonably high for most of the applications. However, higher PSLR is accompanied by a widened beam, i.e., the increased value of minimum possible HPBW.

## 5. Conclusions

In this paper, we extended the idea of staggered beam-steering of subarrays in a semi-coprime array and proposed a mechanism to optimize the extent of staggering (Δ) and side lobe attenuation (SLA) using the Chebyshev weighting technique. We generated, in Matlab, a dataset of numerous combinations of the array parameters and the corresponding performance metrics i.e., HPBW and PSLR. This dataset has been used to train ANN to suggest a suitable combination of the adjustable parameters Δ and SLA, which can promise the minimum HPBW for the given steering angle and PSLR. We have investigated multiple neural networks with a different number of hidden layers and a different number of neurons per layer. Among the investigated network architectures, the best performer was the one with 3 hidden layers with 32 neurons in each layer, which offered MAE within 0.83%. However, another network architecture with only a single hidden layer with 16 neurons offered MAE within 1.82%, which is slightly higher than the best performer.

## Figures and Tables

**Figure 1 sensors-24-07260-f001:**
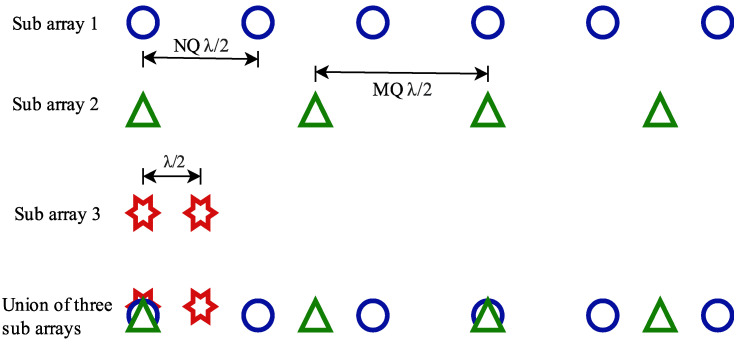
A typical arrangement of SCA with *M* = 3, *N* = 2, *P* = 2, *Q* = 2.

**Figure 2 sensors-24-07260-f002:**
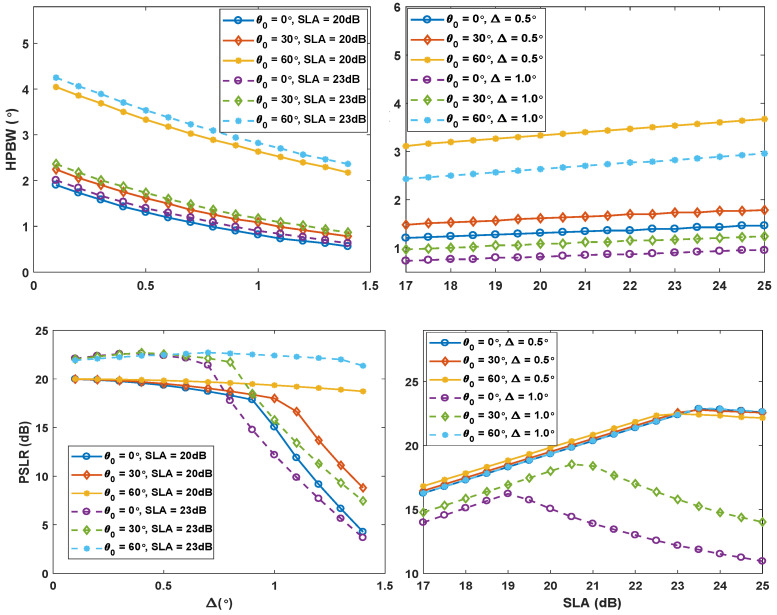
Effect of change in Δ and SLA on HPBW and PSLR for (M,N,P,Q) = (3, 2, 3, 3).

**Figure 3 sensors-24-07260-f003:**
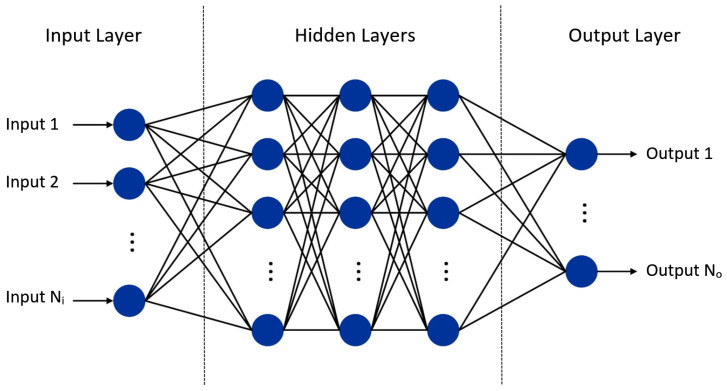
A general architecture of an artificial neural network.

**Figure 4 sensors-24-07260-f004:**
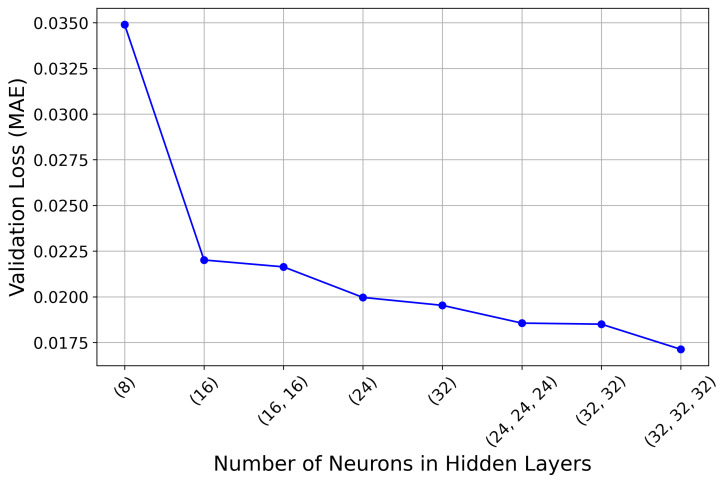
Performance of ANN with different numbers of hidden layers and numbers of neurons per layer.

**Figure 5 sensors-24-07260-f005:**
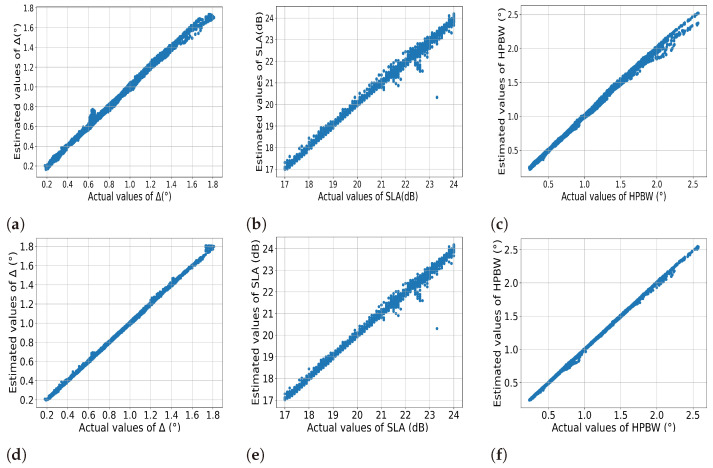
Actual vs Estimated Values (**a**–**c**) ANN1 (**d**–**f**) ANN2.

**Table 1 sensors-24-07260-t001:** Set of Values of the Parameters for the Dataset Generation.

Parameters	Values
(M,N,P,Q)	(3, 2, 3, 3), (4, 3, 3, 2), (5, 2, 3, 3), (6, 5, 3, 2)
$\theta_{0}$	0° to 70°, step size = 0.5°
SLA	17 to 25 dB, step size = 0.1 dB
Δ	0° to 2°, step size = 0.1°

**Table 2 sensors-24-07260-t002:** A sample of raw training examples with PSLR = 21.29 dB for (M,N,P,Q) = (3, 2, 3, 3), $\theta_{0}$ = 14°, 14.5°.

Inputs of the ANN						Outputs of the ANN		
M	N	P	Q	$\theta_{0}\;\left({}^{\circ}\right)$	**PSLR (dB)**	**SLA (dB)**	Δ (°)	**HPBW (**°**)**
3	2	3	3	14	21.29	21.5	0.3	1.701
3	2	3	3	14	21.29	21.75	0.45	1.498
3	2	3	3	14	21.29	22.25	0.65	1.267
3	2	3	3	14	21.29	23.5	0.7	1.253
3	2	3	3	14	21.29	24	0.67	1.295
3	2	3	3	14.5	21.29	21.5	0.3	1.708
3	2	3	3	14.5	21.29	22.25	0.65	1.274

**Table 3 sensors-24-07260-t003:** Neural Network Architectural Parameters.

Network Specifications		ANN1	ANN2
Connectivity Type		Fully connected	
Number of Hidden Layers		1	3
Neurons per Hidden Layer		16	32
Activation	Input/Output Layers	Linear	
Function	Hidden Layers	ReLU	
Loss Function		Mean Absolute Error	

**Table 4 sensors-24-07260-t004:** A sample of the test examples for (M,N,P,Q) = (3, 2, 3, 3).

*M*	*N*	*P*	*Q*	$\theta_{0}\;\left({}^{\circ}\right)$	PSLR (dB)	SLA (dB)	Δ (°)	HPBW (°)
3	2	3	3	5.2	17.4	19.3	0.84	0.938
3	2	3	3	5.2	19.7	21.3	0.81	1.036
3	2	3	3	5.2	22	23.1	0.68	1.211
3	2	3	3	13.15	15.9	17.8	0.84	0.938
3	2	3	3	13.15	22.2	23.2	0.68	1.26
3	2	3	3	28.87	15.5	17.5	0.95	1.015
3	2	3	3	28.87	17.7	19.6	0.96	1.085
3	2	3	3	28.87	21.7	22.8	0.79	1.358
3	2	3	3	37.19	17	18.9	1.05	1.169
3	2	3	3	37.19	19.9	21.5	1.01	1.302
3	2	3	3	37.19	22.4	23.4	0.82	1.561
3	2	3	3	49.21	15.9	17.8	1.26	1.386
3	2	3	3	49.21	18.5	20.3	1.28	1.491
3	2	3	3	49.21	20.9	22.2	1.13	1.722
3	2	3	3	49.21	23	23.9	0.94	1.995
3	2	3	3	58.35	16.9	18.8	1.59	1.771
3	2	3	3	58.35	21.4	22.6	1.36	2.212
3	2	3	3	58.35	23	24	1.17	2.492

**Table 5 sensors-24-07260-t005:** Estimation Error of the Proposed ANNs.

	Mean Absolute Percentage Error		
	**SLA**	Δ	**HPBW**
ANN1	0.21%	1.82%	1.53%
ANN2	0.20%	0.83%	0.70%

## Data Availability

All the data is available within the article.

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
