# Peer review of "ANN-Assisted Beampattern Optimization of Semi-Coprime Array for Beam-Steering Applications"

_sensors, 2024, doi:10.3390/s24227260_

Round 1

Reviewer 1 Report

Comments and Suggestions for Authors

The manuscript proposes an artificial neural network (ANN) to optimize the combination of adjustable parameters given a set of array structural parameters and a desired steering angle and peak side lobe ratio (PSLR). The trained ANN can optimize the values of the adjustable parameters to guarantee the minimum HPBW for a given steering angle, PSLR, and array structural parameters. However, there are some major issues that raise my concerns.

 1. The ANN dataset in this paper is based on MTALAB. Has the influence of array factor pattern or array element type been considered? Has the influence of array element coupling on its results been considered? The establishment of the database needs to consider these practical factors to enhance the persuasiveness of the proposed optimization method.

 2. The ANN optimization method introduced in the paper is only used in narrowband arrays. Is it saleable to broadband arrays? If feasible, what improvements need to be made?

 3. The paper proposes an ANN for semi-coprime array with staggered steering of sub arrays, but the innovation of the paper is not clear enough because the ANN used in the paper is relatively common.

 4. Figure 1 and Table 3 have exceeded the display frame. It is recommended to adjust them appropriately to improve appearance.

 5. In terms of language expression, it is recommended to carefully check the entire paper, especially the use of verb tenses.

 6.The legend in Figure 2 blocks the curves in the figure, which should be improved. In addition, the curves in Figure 5 lack a legend.

Author Response

as PDF file

Reviewer 2 Report

Comments and Suggestions for Authors

This research work has demonstrated the idea of staggered beam steering of sub arrays in a semi coprime array and proposed a mechanism to optimize the extent of staggering (∆) and side lobe attenuation (SLA) using Chebyshev weighting technique. Although the idea is attractive, the authors are encouraged to address the following comments to improve the quality of their work before final decision.

1) The title of this work is confusing, please modify it including the proposed approaches and the possible applications.

2) Please remove the discussion on the background of the work in the abstract section, and only focus on the research aims, the applied techniques and design, its advantages and applications.

3) Please support the abstract section with numerical research achievements.

4) Please add more keywords.

5) The introduction section can be improved by adding proper discussions and references on the array antennas. Below can be helpful suggestions.

"Low-Cost Multiband Four-Port Phased Array Antenna for Sub-6 GHz 5G Applications With Enhanced Gain Methodology in Radio-Over-Fiber Systems Using Modulation Instability", IEEE Access, vol. 12, pp. 117787-117799, 2024.

"On the Performance of a Photonic Reconfigurable Electromagnetic Band Gap Antenna Array for 5G Applications", IEEE Access, vol. 12, pp. 60849-60862, 2024.

"Wideband Endfire Antenna Array for 5G mmWave Mobile Terminals", IEEE Access, vol. 12, pp. 39926-39935, 2024.

6) Please provide more discussions on the typical arrangement of SCA with M=3, N=2, P=2, Q=2 shown in Fig.1?

7) Please elaborate how the authors have obtained the effect of change in ∆ and SLA on HPBW and PSLR for (M, N, P, Q) = (3,2,3,3) which are plotted in Fig.2?

8) The working principle of the general architecture of the artificial neural network shown in Fig.3 should be described in depth.

9) Conclusion should be supported with some numerical research findings.

10) More references can be added to the reference part to improve this section.  

Author Response

as PDF file

Round 2

Reviewer 1 Report

Comments and Suggestions for Authors

no comments

Reviewer 2 Report

Comments and Suggestions for Authors

The required changes and modifications were successfully applied by the authors, so the revised version is satisfactory and there are no more technical comments from this reviewer's point of view.